# Majority of pediatric dengue virus infections in Kenya do not meet 2009 WHO criteria for dengue diagnosis

Aslam Khan[1]*, Bryson Ndenga[2], Francis Mutuku[3], Carren M. Bosire[4], Victoria Okuta[2], Charles O. Ronga[2], Noah K. Mutai[2], Sandra K. Musaki[2], Philip K. Chebii[5], Priscilla W. Maina[5], Zainab Jembe[5], Jael S. Amugongo[5], Said L. Malumbo[5], Charles M. Ng'ang'a[5], Desiree LaBeaud[1]

1 Department of Pediatrics, Division of Infectious Diseases, Stanford University School of Medicine, Stanford, California, United States of America, 2 Centre for Global Health Research, Kenya Medical Research Institute, Kisumu, Kenya, 3 Department of Environment and Health Sciences, Technical University of Mombasa, Mombasa, Kenya, 4 Department of Pure and Applied Sciences, Technical University of Mombasa, Mombasa, Kenya, 5 Vector-Borne Diseases Unit, Msambweni County Referral Hospital, Msambweni, Kwale, Kenya

* akhan1@stanford.edu

**Data Availability Statement:** An anonymized dataset has been uploaded to the DRYAD database. The information for the dataset including DOI is:

## Abstract

From 1975–2009, the WHO guidelines classified symptomatic dengue virus infections as dengue fever, dengue hemorrhagic fever, and dengue shock syndrome. In 2009 the case definition was changed to a clinical classification after concern the original criteria was challenging to apply in resource-limited settings and not inclusive of a substantial proportion of severe dengue cases. Our goal was to examine how well the current WHO definition identified new dengue cases at our febrile surveillance sites in Kenya. Between 2014 and 2019 as part of a child cohort study of febrile illness in our four clinical study sites (Ukunda, Kisumu, Msambweni, Chulaimbo) we identified 369 dengue PCR positive symptomatic cases and characterized whether they met the 2009 revised WHO diagnostic criteria for dengue with and without warning signs and severe dengue. We found 62% of our PCR-confirmed dengue cases did not meet criteria per the guidelines. Our findings also correlate with our experience that dengue disease in children in Kenya is less severe as reported in other parts of the world. Although the 2009 clinical classification has recently been criticized for being overly inclusive and non-specific, our findings suggest the 2009 WHO dengue case definition may miss more than 50% of symptomatic infections in Kenya and may require further modification to include the African experience.

## Introduction

Dengue virus is a flavivirus now widespread throughout the tropical regions of the world and is primarily transmitted by the mosquito vector *Aedes aegypti* [1]. There are four known serotypes that cause variable presentation in humans ranging from no symptoms to severe disease manifesting as hemorrhagic fever and shock [2]. With the variation in disease severity and

Khan, Aslam (2022), Dengue Symptoms , Dryad, Dataset, https://doi.org/10.5061/dryad.wdbrv15pv.

**Funding:** The cohort study was funded through the National Institute of Allergy and Infectious Diseases of the National Institutes of Health [R01 AI102918; PI: DL]. The funders had no role in study design, data collection and analysis, decision to publish, or preparation of this manuscript.

**Competing interests:** The authors have declared that no competing interests exist.

widespread prevalence, there have been multiple efforts to provide comprehensive and effective diagnostic criteria for symptomatic infection to address the clinical, surveillance, and research settings [3–5]. The World Health Organization (WHO) has revised diagnostic criteria for symptomatic infection since 1975 with the most recent update in 2009. The 2009 revised guidelines shifted from dengue fever, dengue hemorrhagic fever, and dengue shock syndrome to a new classification of dengue with/without warning signs and severe dengue, with the goal of including more cases into the case definition and helping clinicians distinguish severe disease (Fig 1) [5, 6]. There has been critique citing overinclusion with the nonspecific symptoms defined in the most recent criteria [7]. Furthermore, with dengue infection a large proportion of the population remains asymptomatic when infected, with estimates up to three quarters of all infections worldwide. Multiple reports have described severe disease in children in the Americas and Southeast Asia and we aimed to characterize known symptomatic disease at our clinical sites in Kenya [8].

## Methods

### Ethics statement

This study was approved by the Institutional Review Boards (IRB) at Stanford University by the Administrative Panel on Human Subjects in Medical Research and at the Technical University of Mombasa by the Office of the TUM Ethical Review Committee. Written informed consent was obtained from parents/guardians of all participants.

This research was funded by the National Institute of Allergy and Infectious Diseases of the National Institutes of Health [R01 AI102918; PI: ADL]. We followed children presenting with acute undifferentiated febrile illness in four outpatient clinical study sites across Kenya from January 2014 –June 2019. Febrile participants were enrolled from Chulaimbo Sub-County Hospital and Mbaka Oromo Dispensary (Chulaimbo, a rural Western setting), Jaramogi Oginga Odinga Teaching & Referral Hospital (Kisumu, an urban Western setting), Msambweni County Referral Hospital (Msambweni, a rural coastal setting), and Diani Health Centre (Ukunda, an urban coastal setting). All four hospitals are operated by the Kenyan Ministry of Health.

We recruited all children 1–17 years old who presented for acute febrile illness defined as reported illness during the prior 14 days and current temperature >38˚C with no localizing symptoms. Participants or their parents/guardians provided consent. Our clinical officers obtained detailed clinical histories, comprehensive demographic/household data, and performed physical examinations. All study participants underwent blood collection for serologic analysis, molecular testing, and malaria parasite smear. All samples were tested by reverse transcriptase polymerase chain reaction (RT-PCR) testing for dengue virus serotypes 1–4 and chikungunya virus [9, 10]. RT-PCR testing was performed at our field laboratory sites in Msambweni and Kisumu and reproduced at Stanford University with a pan-DENV real time RT-PCR [10, 11]. All information was uploaded via a secure server into RedCap accessible at Stanford University.

We evaluated symptoms for all positive dengue virus individuals by RT-PCR as described by the clinical officers in their physical examinations. There was limited additional laboratory studies available but all available data was collected in reference to the 2009 WHO criteria for dengue diagnosis (Fig 1). Participants were classified as dengue infection without warning signs, dengue infection with warning signs, severe dengue, and did not meet criteria. Dengue was defined as a combination of ≥2 clinical findings in the febrile individuals, including nausea, vomiting, rash, aches and pains, positive tourniquet test, and leukopenia. Warning signs included abdominal pain/tenderness, persistent vomiting, clinical fluid accumulation, mucosal

**Fig 1. World Health Organization (WHO) dengue classification (2009).**

bleeding, lethargy, restlessness, and hepatomegaly. Severe dengue was defined by dengue with any of the following: severe plasma leakage leading to shock or fluid accumulation with respiratory distress, severe bleeding, or severe organ impairment (elevated transaminases $\geq$1,000 IU/L, impaired consciousness, or heart impairment).

The participants were classified by disease severity with the cases not meeting criteria grouped as a separate category. Proportions were calculated per group with the denominator of all participants included in the study (N = 369). For each proportion we calculated the exact binomial confidence interval (95%). The mean ages were calculated by disease group and symptoms were tallied by disease group.

## Results

Of the 369 identified cases at our study sites, 7% (27/369, 95% CI [4.9–10.5%]) met criteria for dengue without warning signs, 29% (108/369, 95% CI [24.7–34.2%]) met criteria for dengue with warning signs, 2% (7/369, 95% CI [0.8–3.9%]) met criteria for severe dengue, and 62% (227/369, 95% CI [56.3–66.5%]) did not meet criteria for diagnosis. The mean ages were 5.0, 4.5, 4.7, and 5.5 years, respectively. The most common warning signs were lethargy paired with aches/pains and nausea/vomiting. Table 1 demonstrates the percentages of symptoms paired with the respective diagnoses. For severe dengue (N = 7) all cases had impaired consciousness but none were notable for hemorrhagic symptoms. No patients had laboratory data collected to determine white blood cell count, platelet count, or liver transaminase levels.

## Discussion

This study highlights the challenges with dengue diagnosis and the current clinical criteria in Kenya. The majority of the symptomatic cases did not meet criteria per the 2009 guidelines, differing from other experiences reported in the Americas and Asia, where disease severity

**Table 1. Symptoms of participants (n = 369) characterizing dengue disease.** Abdominal pain/tenderness and lethargy are both warning signs.

|  | Aches/Pains | Nausea/Vomiting | Rash | Abdominal Pain/Tenderness | Lethargy | Impaired Consciousness |
|---|---|---|---|---|---|---|
| Dengue without warning signs (n = 27) | 93% (25) | 81% (22) | 19% (5) | - | - | - |
| Dengue with warning signs (n = 108) | 76% (82) | 57% (62) | 13% (14) | 29% (31) | 80% (86) | - |
| Severe dengue (n = 7) | 86% (6) | 43% (3) | 14% (1) | - | 86% (6) | 100% (7) |
| Did not meet criteria (n = 227) | 38% (86) | 15% (33) | 1% (3) | 8% (19) | 5% (12) | - |

was classified in most symptomatic cases, with severe dengue representing up to 28% of cases [12–17]. A recent systematic review identified a cumulative frequency of plasma leakage which confers severe dengue at 36.8% which contrasts to this cohort where no hemorrhagic symptoms were identified. Analyzing the utility of the guidelines from a clinical perspective, the few participants with severe disease were identified in the cohort. However, from an epidemiologic perspective, there was low utility in application of the guidelines for surveillance purposes as majority of infections did not meet criteria, contrasting to experience from the Americas or Asia [6, 17, 18].

Although our findings differ from other experiences around the world, they are similar to other experiences reported from the African continent [19–22]. With improved surveillance, dengue has been increasingly identified as a significant cause of acute febrile illness across various sub-Saharan African countries [11, 19, 21]. The most common symptom often reported is fever with headaches, aches/pains, or myalgias, as demonstrated with our study (Table 1) [11, 19, 22]. This study evaluated 369 Kenyan children and did not identify any hemorrhagic symptoms or leaky capillary syndrome as described in children in Asia or the Americas [12, 14]. All the severe dengue cases met criteria with impaired consciousness which could be from infection or other causes. Co-infection is possible with other pathogens like malaria and comorbidities like epilepsy can also alter the presentation [11, 23, 24].

There can be multiple explanations why the majority of dengue infections were not captured within the confines of the guidelines. First, the majority of dengue infections cause mild illness or no symptoms at all, consistent with our findings [11, 22, 25]. Although the participants in our cohort did develop undifferentiated fever and sought clinical care, suggesting an inflammatory response significant enough to enter the health centers, the majority of their examinations were not concerning for severe features [11, 25]. Second, clinician subjectivity during examination may discount severe features [26]. As there are many possible causes of undifferentiated fever in Kenya, including malaria which is often the most suspected pathogen, clinicians rarely consider other causes of illness as they be unfamiliar with laboratory confirmed disease in the setting of limited diagnostics [27]. In our study the clinical officers had dengue PCR available and were trained on dengue diagnosis, which is not the case in many other clinical settings. Additionally, the health centers are all affiliated with the Ministry of Health and often there is a very high daily volume of patients, presenting a burden to be as comprehensive as one would strive to achieve. Third, there are reports of minimal severity of dengue disease in individuals of African ancestry when compared to white individuals [20]. This phenomenon has been demonstrated in Cuba and Haiti during prior outbreaks, leading experts to believe there may be a protective genetic factor preventing severe disease manifestations [19]. This theory has been proposed to explain the differences in the African dengue experience with the rest of the world [19].

The limitations of this study include the variability of clinical officer practices and limited laboratory data available. There was no significant laboratory data to evaluate the complete blood cell count parameters or hepatic involvement of all participants which may have led to misclassification bias amongst our cohort. In recent outbreaks, leukopenia has been reported in as much as 66% of symptomatic cases and if applied at that rate to this cohort, 47% of cases would continue to not meet criteria for dengue diagnosis [28–30].

With minimal hemorrhagic symptoms and most infections not meeting criteria in the real-world limited laboratory setting, there may be benefit to modify the criteria to account for wide variability of presentation. The modification may need to be stratified for differing purposes like surveillance versus clinical diagnosis. As demonstrated in this study, there is wide variability with dengue infection and a single set of criteria may not be sufficient for all purposes. Ultimately, although there were less severe symptoms identified in our pediatric cohort

in Kenya, it is important to continue surveillance efforts and improving diagnosis of dengue infection to detect and ultimately prevent future outbreaks.

## Author Contributions

**Conceptualization:** Aslam Khan, Desiree LaBeaud.

**Data curation:** Carren M. Bosire, Victoria Okuta, Charles O. Ronga, Noah K. Mutai, Sandra K. Musaki, Zainab Jembe, Jael S. Amugongo, Said L. Malumbo, Charles M. Ng'ang'a, Desiree LaBeaud.

**Formal analysis:** Aslam Khan, Philip K. Chebii.

**Funding acquisition:** Desiree LaBeaud.

**Investigation:** Bryson Ndenga, Francis Mutuku, Carren M. Bosire, Victoria Okuta, Charles O. Ronga, Noah K. Mutai, Sandra K. Musaki, Philip K. Chebii, Priscilla W. Maina, Zainab Jembe, Jael S. Amugongo, Said L. Malumbo, Charles M. Ng'ang'a, Desiree LaBeaud.

**Methodology:** Aslam Khan, Desiree LaBeaud.

**Supervision:** Bryson Ndenga, Francis Mutuku, Desiree LaBeaud.

**Writing – original draft:** Aslam Khan.

**Writing – review & editing:** Aslam Khan, Bryson Ndenga, Francis Mutuku, Desiree LaBeaud.

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
