## [Decision Letter · Decision Letter 0]

9 Nov 2021

PGPH-D-21-00576

Majority of Pediatric Dengue Virus infections in Kenya Do Not Meet 2009 WHO Criteria For Dengue Diagnosis

Dear Dr. Khan,

Thank you for submitting your manuscript to PLOS Global Public Health. After careful consideration, we feel that it has merit but does not fully meet PLOS Global Public Health’s publication criteria as it currently stands. Therefore, we invite you to submit a revised version of the manuscript that addresses the points raised during the review process.

We look forward to receiving your revised manuscript.

Kind regards,

Max Carlos Ramírez-Soto, BSc, MPH

Academic Editor

Journal Requirements:

1. Thank you for including your ethics statement:  "This study was approved by the Institutional Review Boards (IRB) at Stanford University and at our affiliated sites in Kenya.".   

For additional information about PLOS Global Public Health ethical requirements for human subjects research, please refer to http://journals.plos.org/globalpublichealth/s/submission-guidelines#loc-human-subjects-research.

2. Figure 1 appears to have been [previously published. We do not recommend using standard request forms available on Publishers' websites, as they grant single use rather than republication under an open access license. Instead, we recommend replacing the image. 

3. We ask that a manuscript source file is provided at Revision. Please upload your manuscript file as a .doc, .docx, .rtf or .tex. If you are providing a .tex file, please upload it under the item type ‘LaTeX Source File’ and leave your .pdf version as the item type ‘Manuscript’.

4. Tables should not be uploaded as individual files.  Please remove these files and include the tables in your manuscript file.

5. In the online submission form, you indicated that your data will be submitted to the Dryad database upon acceptance. Should your submission be accepted, we will require the following information in your Data Availability Statement: 

1. The DOI provided by Dryad

2. The citation for your data package in the reference section of your manuscript

3. The citation for your data package in the methods section

If you are unable to adhere to our open data policy, please kindly revise your statement to explain your reasoning and we will seek the editor's input on an exemption. Please be assured that, once you have provided your new statement, the assessment of your exemption will not hold up the peer review process.

Reviewers' comments:

Reviewer's Responses to Questions

**Comments to the Author**

1. Does this manuscript meet PLOS Global Public Health’s publication criteria? Is the manuscript technically sound, and do the data support the conclusions? The manuscript must describe methodologically and ethically rigorous research with conclusions that are appropriately drawn based on the data presented.

Reviewer #1: Yes

Reviewer #2: Yes

Reviewer #3: Partly

2. Has the statistical analysis been performed appropriately and rigorously?

Reviewer #1: Yes

Reviewer #2: No

Reviewer #3: N/A

3. Have the authors made all data underlying the findings in their manuscript fully available (please refer to the Data Availability Statement at the start of the manuscript PDF file)?

Reviewer #1: Yes

Reviewer #2: Yes

Reviewer #3: No

4. Is the manuscript presented in an intelligible fashion and written in standard English?

Reviewer #1: Yes

Reviewer #2: Yes

Reviewer #3: Yes

5. Review Comments to the Author

Reviewer #1: I think it would be important to compare the absolute numbers and

percentages of the findings of this study with respect to the classification criteria of the WHO, with other studies and include them in the discussion. They could also expand the information if the cases were obtained in times of greater or lesser dengue transmission and in the future increase the sample size

Reviewer #2: the statistical analysis is not complete as the "p-value" and the confidence Interval have not been calculated.

1.1. Summary of the research

The authors investigated the ability of health workers to diagnose new cases of Dengue based on the 2009 WHO revised guidelines.

To do so, they conducted a prospective cohort study from 2014 to 2019 in 4 health facilities in Kenya. They first tested patients admitted with fever in these facilities for RT-PCR and then included all Dengue positive cases, 369 cases. The next step was to search these 369 cases for the WHO diagnostic criteria according to the 2009 classification of new Dengue cases.

The results showed that 62% of the RT-PCR positive Dengue cases did not meet the WHO diagnostic criteria. The study also showed that severe Dengue cases were less frequent in Kenya compared to American or Asian series.

1.2. overall impression

This is an excellent, well-written, well-presented article that meets the standards of a scientific paper. In addition, the research question is relevant, and the results found are very useful for health professionals to improve the diagnosis and management of new cases of Dengue. Indeed, the diagnosis of Dengue is difficult because the non-specific symptoms and the frustrated forms represent 80 to 90% of the cases. However, its evolution is unpredictable while the lethality of severe forms, which is 1 to 5% of cases on average, can reach 20% in the absence of early and adapted medical care.

Also, the type of epidemiological study chosen, a prospective cohort study, is of satisfactory power.

There are some limitations to the study, which if corrected could improve the quality

Strengths

- The research question is relevant

- The authors have done a good literature review and shared all the references used to conduct their work with a good indexing of the articles (see page 12)

- The introduction clearly states the research question and the conclusion answers this question in an objective, factual manner (see page 5).

- Good research methodology: the authors conducted a prospective cohort study, the sample size is sufficient (>300) for the results to be significant. Written consent was sought from the parents of the study participants, and the study protocol was submitted to the ethics committee (see pages 5 to 7).

- The authors looked for certain co-morbidities such as Chikungunya and malaria, since in tropical areas, these are the frequent and serious diseases to be suspected in front of an isolated fever.

Weaknesses

- The absence of biological test in the research protocol is a problem knowing that Dengue symptoms are not very specific and that biological criteria are part of the classification of new cases of Dengue, both simple and severe. (see page 7 and Figure 1 page 11 of the manuscript)

- During the analysis of the results, the "p-value" was not calculated, nor the confidence intervals, not allowing to conclude that the results were not due to chance alone (see page 11).

- The RT-PCR test should also be used to search for other arboviruses besides Chikungunya (Zika virus, Yellow fever, Rift Valley fever, Crimean-Congo virus, etc...) as there are cross-reactions between them which can lead to false positive cases of Dengue. (see page 6).

Recommendations

- Revise the 2009 WHO guidelines to allow for better and earlier diagnosis of Dengue fever cases and thus prevent complications

- Systematically search for Dengue in all cases of isolated fever in endemic areas according to the results of the study.

- Propose a classification of Dengue cases specific to Africans who have a different epidemiological profile than Americans and Asians

2. Discussion of specific areas for improvement

-we suggest that in the "Results" section on page 8 and page 11 of the manuscript, the authors may complete the data analysis by calculating the "p-value" and the confidence intervals.

- If the samples are still available, may the authors do the testing for other arboviruses in order to be sure that the RT-PCR is positive for the Dengue Virus for the cases that not meet the WHO criteria.

Reviewer #3: The paper does not present enough evidence to support its single most overriding communication objective, which is that WHO revised case definition is not sensitive for Dengue fever. The case definition for probable dengue includes two of five symptoms, one of which (leucopenia) was never assessed in all cases. It is surprising however that children numbering as much who are visiting medical facilities including reference hospitals with febrile illness do not have records of blood counts. The methods section did not offer a full description of the diagnostic process. For example, no information was given as to the name, description or sources of the primers/kits used in the polymerase chain reaction performed in the field sites and at Stanford University. Such would offer opportunity for appraisal of the quality of these primers and their capacity to prevent cross-reactivity with other febrile illness causing viruses. Data as presented in table 1 is not clear and no reference to this table was made in the discussion that can help clarify what figures and percentages represent.

6. PLOS authors have the option to publish the peer review history of their article (what does this mean?). If published, this will include your full peer review and any attached files.

**Do you want your identity to be public for this peer review?** For information about this choice, including consent withdrawal, please see our Privacy Policy.

Reviewer #1: No

Reviewer #2: No

Reviewer #3: No

---

## [Editor Report · Decision Letter 1]

9 Mar 2022

Majority of Pediatric Dengue Virus infections in Kenya Do Not Meet 2009 WHO Criteria For Dengue Diagnosis

PGPH-D-21-00576R1

Dear Dr. Khan,

We are pleased to inform you that your manuscript 'Majority of Pediatric Dengue Virus infections in Kenya Do Not Meet 2009 WHO Criteria For Dengue Diagnosis' has been provisionally accepted for publication in PLOS Global Public Health.

Best regards,

Max Carlos Ramírez-Soto, BSc, MPH, FRSPH

Academic Editor